# Development of a brief scoring system to predict any-cause mortality in patients hospitalized with COVID-19 infection

Nasheena Jiwa [1,2] *, Rahul Mutneja[2], Lucie Henry[3], Garrett Fiscus[3], Richard Zu Wallack[2]

**1** Department of Pulmonary and Critical Care Medicine, University of Connecticut Health Center, Farmington, CT, United States of America, **2** Department of Pulmonary and Critical Care Medicine, St. Francis Hospital, Hartford, CT, United States of America, **3** Department of Internal Medicine, University of Connecticut Health Center, Farmington, CT, United States of America

* jiwa@uchc.edu

**Data Availability Statement:** All data that is available is incorporated in the manuscript and attached figures. There are no other documents for supporting data that are available.

## Abstract

Patients hospitalized with COVID-19 infection are at a high general risk for in-hospital mortality. A simple and easy-to-use model for predicting mortality based on data readily available to clinicians in the first 24 hours of hospital admission might be useful in directing scarce medical and personnel resources toward those patients at greater risk of dying. With this goal in mind, we evaluated factors predictive of in-hospital mortality in a random sample of 100 patients (derivation cohort) hospitalized for COVID-19 at our institution in April and May, 2020 and created potential models to test in a second random sample of 148 patients (validation cohort) hospitalized for the same disease over the same time period in the same institution. Two models (Model A: two variables, presence of pneumonia and ischemia); (Model B: three variables, age > 65 years, supplemental oxygen $\geq$ 4 L/min, and C-reactive protein (CRP) > 10 mg/L) were selected and tested in the validation cohort. Model B appeared the better of the two, with an AUC in receiver operating characteristic curve analysis of 0.74 versus 0.65 in Model A, but the AUC differences were not significant (p = 0.24. Model B also appeared to have a more robust separation of mortality between the lowest (none of the three variables present) and highest (all three variables present) scores at 0% and 71%, respectively. These brief scoring systems may prove to be useful to clinicians in assigning mortality risk in hospitalized patients.

## Introduction

The COVID-19 pandemic peaked in Connecticut and other states in the Northeast in April and May, 2020 [1], resulting in dramatic increases in hospitalizations and mortality, and putting considerable stress on health care systems. One striking feature of this novel coronavirus disease is its variability in clinical outcome, ranging from inapparent infection to prolonged hospitalization and death. However, for those individuals with sufficient disease burden and co-morbid conditions to warrant hospitalization, mortality risk is high, although this too varies widely among health care systems [2–5]. The availability of a simple and brief method of

**Funding:** The authors received no specific funding for this work.

**Competing interests:** The authors have declared that no competing interests exist.

mortality risk prognostication early on in the hospitalization could focus specific COVID-19 therapies and direct scarce personnel and therapeutic resources to those at greatest risk. Our prediction model is intended to be a used as a tool for quick interpretation of patient data within the first 24 hours, to predict mortality outcomes. Accordingly, we sought to develop and test a simple scoring system based on clinical factors and laboratory tests frequently ordered within the first 24 hours of admission that could reasonably predict mortality in those hospitalized with COVID-19 infection.

## Methods

Our retrospective study had two components: 1) An initial review of records from adults hospitalized with a clinical diagnosis of COVID-19 infection, evaluating demographic, socioeconomic, and clinical factors as predictors of in-hospital mortality, with a goal of positing one or more brief prognostic scoring systems (derivation cohort); and 2) Testing the proposed scoring system(s) by using data from a second review of hospitalized patients (validation cohort). Our intent was to create a prognostic tool that was brief and simple to administer, yet sufficiently predictive of mortality to be useful to health care professionals. The Trinity Health of New England Institutional Review Board granted approval prior to study initiation. Patient data was collected in a coded and encrypted fashion. The IRB waived the requirement for informed consent as this was a retrospective study.

For the first component of the study, we reviewed in-hospital records from a randomized sample of 100 COVID-19 patients (out of approximately 1600 records), who had been admitted to our tertiary-care center in Hartford, CT, during the months of April and May, 2020. These patients are known as the derivation cohort. All patients had been admitted and hospitalized with a clinical diagnosis and serological confirmation of COVID-19 infection.

The choice of variables abstracted from electronic hospital records in this initial review was based on negative prognostic factors that were available in peer-reviewed medical literature at the time of the review [6, 7] and on our clinical judgment. These included:

1. Demographics: age, gender, race-ethnicity. For the latter analysis, self-reported Hispanic was categorized separately from self-reported Asian, Black and Caucasian groups in the medical record

2. Socioeconomic status (SES): Our surrogate marker for low SES was Medicaid dual Medicare-Medicaid or no-insurance status. (Medicaid refers to health coverage for those with very low income, dual Medicare-Medicaid refers to health coverage for those above the age of 65, or under the age of 65 with a disability and low income status).

3. Referral source: home versus extended care facility (ECF)

4. Clinical abnormalities within the first 24 hours: fever, hypoxemia, including oxygen therapy, supplemental oxygen requirement in liters/minute

5. Need for supplemental high-flow oxygen therapy in the first 24 hours, as defined as a flow rate $\geq$ 4 L/min or the need for high-flow oxygen therapy

6. Recorded presence of co-morbidity, as documented by the admission history: hypertension, diabetes, obesity, COPD, asthma, chronic liver disease, deep venous thrombosis, atrial fibrillation, coronary artery disease, chronic kidney disease, history of cerebrovascular disease, history of congestive heart failure, history of malignancy

7. Treatment with angiotensin converting enzyme inhibitors or blockers (ACE or ARBs)

8. Initial laboratory tests: CBC, general chemistries (including creatinine), troponin, b-type natriuretic protein, C-reactive protein

9. Radiographic abnormality: consolidation on chest x-ray or CT [8].

10. The presence of ischemia in the first 24 hours of admission, defined as an elevated troponin level (of $> 0.04$ pg/L)

In-patient pharmacologic treatments and results from laboratory tests that were administered or preformed pre-hospitalization (such as Vitamin D levels or radiographs) were not analyzed.

A second sample of different patients called the validation cohort of patients was then reviewed. These records were randomly selected from the same population of hospitalized patients in April and May, 2020 (excluding those in the original sample), until an arbitrary number of at least 40 in-hospital deaths from COVID-19 were reviewed. Those variables found predictive of mortality in the univariate analyses in the original sample were abstracted.

## Statistical analysis

We then used univariate analyses (proc Logistic, SAS version 9.4) to determine which potential explanatory variables significantly predicted any-cause, in-hospital mortality from COVID-19. For these analyses we created predictive models using 2-category dichotomous adaptations of these variables (0 = absent, 1 = present) and then, using an iterative approach, we then tested the most robust models as predictors of mortality in a second, independent sample of 148 hospitalized COVID patients over the same time period in the same institution. The goal was to identify a brief and easy-to-use model that had the highest area under the curve (AUC) in Receiver Operating Characteristic (ROC) curve analyses [9, 10] and had good separation in mortality between highest (i.e., worst) and lowest scores. For reference, an AUC between 0.7 and 0.8 is considered acceptable, while an AUC between 0.8 and 0.9 is considered excellent [11]. ROC curves were compared using the SAS logistic procedure and the ROCCONTRAST statement. Our study was not designed to analyze post-hospitalization mortality. All living patients had been discharged from the hospital by the time of the analyses.

## Results

### Derivation cohort

In the initial convenience sample of 100 patients, 44% were female, mean (± standard deviation, SD) age was 68 ± 17 years; 52% had low SES, and race/ethnicity was as follows: Asian 5, Black 28, Caucasian 42, and Hispanic 25. Sixty-one percent were older than 65 years; 48% were treated with oxygen $\geq$ 4 Liters/minute within the first 24 hours of their hospital stay; radiographic pneumonia was present in 61%; ischemia in 37%; CRP $> 10$ mg/L was present in 47%; hospital length of stay was 12 ± 16 days; and in-hospital mortality was 36%. The mean number of co-morbid conditions was 2.5 ± 1.7; percentages (in parentheses) were as follows: hypertension (72), insulin-dependent diabetes (23), obesity (52), COPD (12), asthma (14), chronic liver disease (1), history of deep venous thrombosis (7), atrial fibrillation (13), coronary artery disease (23), chronic kidney disease (18), history of cerebrovascular disease (15), history of heart failure (23), history of malignancy (4). For some variables fewer than 100 data values were available; among these were (available numbers for analysis in parentheses): race/ethnicity (81), SES (98), ischemia (99), pneumonia (98), CRP (95), BNP (42).

## Validation cohort

In this second sample of 148 subjects, 44% were female; age was 69 ± 14 years, 64% had low SES, 2% were Asian, 43% Black, 41% Caucasian, and 14% Hispanic. Fifty-nine percent were older than 65 years; 47% were treated with supplemental oxygen $\geq$ 4 Liters/minute; radiographic pneumonia was present in 64%; ischemia in 47%; CRP was elevated $>$ 10 mg/L in 49%; hospital length of stay was 9.0 ± 9.8 days; 45 patients died, giving an in-hospital mortality of 30%. For some variables fewer than 148 data values were available: among these were (available numbers for analysis in parentheses): race/ethnicity (145), SES (147), ischemia (110), pneumonia (142), CRP (130), BNP (61).

Table 1 shows selected patients characteristics for Derivation and Validation Cohorts.

**Selection of predictors and univariate and multivariate logistic regression analysis.**
The following variables were not related to mortality in univariate analyses in the derivation cohort: gender; fever upon admission; respiratory rate; any of the following preexisting conditions: hypertension, CAD, liver disease, non-insulin-dependent diabetes, DVT, history of CVA, chronic kidney disease, history of CHF, history of malignancy, COPD, asthma, interstitial lung disease; prescribed ACE or ARB; or any of the following laboratory data: lactic acid, hemoglobin, total WBC, lymphocyte count, platelet count, BNP, procalcitonin, ferritin, d-dimer, albumin, bilirubin, or creatinine.

Caucasians were older and tended to have a higher mortality than Blacks (42% versus 30%), but the difference was not statistically significant (p = 0.36); Hispanic mortality was 40% and also not significantly different from Caucasians or Blacks. Low SES was not a significant predictor of mortality, and residence prior to hospitalization (home versus ECF) was also not significantly predictive.

The following variables were predictive of mortality in the univariate analyses: age, atrial fibrillation, insulin-dependent diabetes (IDDM), C-reactive protein (CRP), supplemental oxygen requirement, pneumonia on initial radiology, and ischemia (Table 2).

Each of the above predictive variables present in the first 24 hours of hospitalization was assigned a 1 or 0 value for the purpose of creating a useable scoring system based on categorical values. These assignments were: age ($<$ 65 years = 0, over 65 years = 1); atrial fibrillation (absent = 0, present = 1); IDDM (absent = 0, present = 1); CRP ($\leq$ 10 mg/L = 0, $>$ 10 mg/L = 1); supplemental oxygen (0–4 LPM = 0, 4+ LPM or high flow O2 therapy = 1); pneumonia

**Table 1. Patient characteristics.**

| Variable | Derivation Cohort (n = 100) | Validation Cohort (n = 148) |
|---|---|---|
| Female (%) | 44 | 44 |
| Age > 65 years (%) | 61 | 59 |
| Race A/B/C/H (%) | 5/28/42/25 | 2/43/41/14 |
| Low SES (%) | 52 | 64 |
| From ECF (%) | 33 | 35 |
| Hospital LOS (days ± SD) | 12 ± 16 | 9 ± 10 |
| Mortality (%) | 36 | 30 |
| Pneumonia (%) | 62 | 64 |
| Ischemia (%) | 37 | 47 |
| CRP > 10 (%) | 47 | 49 |
| High O2 requirement | 44 | 39 |

Race: A: Asian, B: Black, C: Caucasian; H: Hispanic; ECF: extended care facility; LOS: length of stay; CRP = C-reactive protein; O2 = oxygen.

**Table 2. Variables predictive of in-hospital, any-cause mortality in univariate testing in the Derivation Cohort of patients.**

| Variable | OR (95% CI) | p. |
|---|---|---|
| Age (per year) | 1.04 (1.01 to 1.07) | 0.009 |
| Atrial fibrillation (N vs. Y) | 0.20 (0.06 to 0.71) | 0.01 |
| Ischemia (N vs. Y) | 0.36 (0.15 to 0.84) | 0.02 |
| IDDM (N vs. Y) | 0.33 (0.13 to 0.85) | 0.02 |
| CRP (per Δ 1 mg/L) | 1.05 (1.00 to 1.09) | 0.04 |
| Pneumonia (N vs. Y) | 0.15 (0.05 to 0.44) | 0.0005 |
| High O2 req. (N vs. Y) | 0.33 (0.14 To 0.78) | 0.01 |

OR: Odds ratio; CI: 05% confidence interval; N vs. Y: not present versus present; IDDM: insulin-dependent diabetes mellitus; CRP: c-reactive protein; L = liter; O2: oxygen; req.: requirement.

(absent = 0, present = 1); ischemia, reported on the clinical record or with an elevated troponin (> 0.4 ng/mL) (absent = 0, present = 1).

The above categorical variables were entered into a multivariate logistic regression (SAS) in an iterative fashion with a goal of creating a robust mortality prognostic scoring model that was brief, had high AUC in logistic regression, and provided significant separation from the highest score (highest risk, all negative predictor variables present) from lesser scores.

One iteration, Model A, the combination of two variables—pneumonia and ischemia–yielding potential composite scores of 0 (neither variable positive, 25%), 1 (one of the two variables positive, 51%) or 2 (both variables positive, 24%) (Incomplete data on 2 patients), had the highest AUC of 0.74 (95% CI 0.65 to 0.82). Odds ratios for this analysis are in Table 3 while mortality in each score category is given in Table 4.

A second iteration, Model B, combining three variables–age > 65 years, high supplemental oxygen requirement over the first 24 hours of hospitalization, and a CRP > 10 (range of scores 0 to 3)–had the second highest AUC (0.66 (95% CI 0.56 to 0.77). The percentage of patients in Model B was score 0 (20%), score 1 (32%), score 2 (23%) and score 3(25%). Five patients without CRP measurements could not be included in the multivariate analysis. Odds ratios for this analysis are in Table 3 and mortality for each score category is given in Table 4.

**Table 3. Mortality risk by model.**

| | Derivation Cohort | Validation Cohort |
|---|---|---|
| **Model A** | | |
| Score | OR, 95% CI | OR, 95% CI |
| 0 vs. 2 | 0.03 (0.00 to 0.23) | 0.21 (0.58 to 0.75) |
| 1 vs. 2 | 0.40 (0.15 to 1.09) | 0.36 (0.15 to 0.89) |
| **Model B** | | |
| Score | Derivation Cohort | Validation Cohort |
| 0 vs. 3 | 0.11 (0.03 to 0.50) | * |
| 1 vs. 3 | 0.35 (0.11 to 1.06) | 0.10 (0.03 to 0.37) |
| 2 vs. 3 | 0.230 (0.06 to 0.79) | 0.29 (0.09 to 0.95) |

OR: odds ratio; CI: confidence interval

* = no patient in Sample 2 Model B with a score of 0 died.

Model A: pneumonia, ischemia: one point each if present; scores can range from 0–2.

Model B: age > 65, high supplemental oxygen requirement, CRP > 10 mg/L: one point each if present; scores can range from 0–3.

**Table 4. Mortality by score.**

Model A

| Score | Derivation Cohort | Validation Cohort |
|---|---|---|
| | Mortality (%) | Mortality (%) |
| 0 | 4 | 19 |
| 1 | 40 | 29 |
| 2 | 63 | 53 |

Model B

| Score | Derivation Cohort | Validation Cohort |
|---|---|---|
| | Mortality (%) | Mortality (%) |
| 0 | 16 | 0 |
| 1 | 37 | 20 |
| 2 | 27 | 41 |
| 3 | 62 | 71 |

Differences between means (p.) of category scores.

Model A, Derivation Cohort: 0 vs. 1: 0.04; 0 vs. 2: < 0.0001; 1 vs. 2: 0.04.

Model A, Validation Cohort: 0 vs. 1: 0.42; 0 vs. 2: 0.01; 1 vs. 2: 0.02.

Model B, Derivation Cohort: 0 vs. 1: 0.13; 0 vs. 2: 0.43; 0 vs. 3: 0.001; 1 vs. 2: 0,47; 1 vs. 3: 0.04; 2 vs. 3: 0.01.

Model B, Validation Cohort: 0 vs. 1: 0.09; 0 vs. 2: 0.0002; 0 vs 3: < 0.0001; 1 vs. 2: 0.02; 1 vs. 3: < 0.0001; 2 vs. 3: 0.01.

The above two ROC curves in Sample 1 were not significantly different: p = 0.24.

**Validation of the scoring systems.** Both models were tested as predictors of in-hospital mortality in the validation cohort. In this analysis, Model A (two variables: ischemia and consolidation) had a lower AUC than Model B (3 variables: older age, high supplemental oxygen requirement and elevated CRP): 0.65 (95% CI 0.55 to 0.76) versus 0.74 (95% CI 0.65 to 0.83), respectively, but their difference was not significant: p = 0.18. ROC curves for Sample 1 and 2 are given in Fig 1. Of note, only 107 patients could be analyzed in the logistic regression in Model A, mainly from non-existing data on troponin and no clinical diagnosis of ischemia over the first 24 hours of hospitalization. Odds ratios for each score category for both models are in Table 3 and mortality in each score category is given in Table 4.

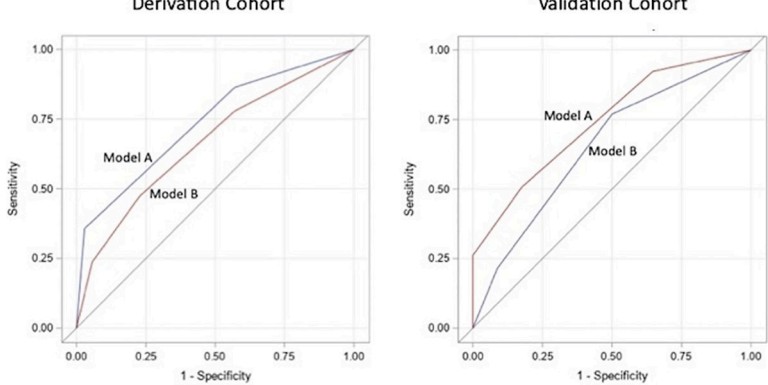

**Fig 1. ROC curves in the derivation and validation cohorts.** Model A: Scoring based on two variables: presence of pneumonia, ischemia; score can range from 0–2. Model B: Scoring based on three variables: age over 65, high supplemental oxygen requirement, C-reactive protein over 10. Presence of each given a score of 1; score can range from 0–3.

## Discussion

Our study provides two simple and easy-to-complete prognostic scoring models based on data readily available in the first 24 hours of hospitalization to predict in-hospital mortality of COVID-19 patients. Model A was based on the presence of ischemia and pneumonia; Model B was based on age > 65 years, high supplemental oxygen requirement, and elevated CRP values. Based on performance in the validation cohort, Model B had a slightly higher AUC (0.74 vs. 0.65), although the difference between the two models was not statistically significant. Model B also tended to perform better with respect to mortality separation in logistic regression. Given the small numbers of subjects in our study, the lack of a statistically significant difference in AUC, and the fact that CRP (a component of model B) was not obtained in all patients upon admission, a strong inference on the comparative performance of the two scoring systems would be problematic.

Model B scoring was especially powerful in separating score category 0 from category 3: those patients without any of the three negative factors had no in-hospital mortality while those with all three had 71% mortality. Intermediate scores had intermediate risk, although the categories were not statistically different in some instances. However, mortality in score category 3 was significantly greater than any of the lesser scores, attesting to the potential usefulness of this category.

While the clinical consequences of using either of these two predictive scoring systems is not determined, information from a simple model such as either of these may provide useful prognostic information to Emergency Department and admitting clinicians, thereby potentially directing scarce personnel and medical resources toward those hospitalized individuals at greatest risk of dying.

Similar to our study, advanced age, elevated levels of CRP, and oxygenation status (either from estimates of oxygenation from pulse oximetry or from supplemental oxygen requirements in our study) were predictive of in-hospital mortality in other analyses [8, 12–15]. However, in contrast, we were not able to demonstrate that sex, obesity [14] or D-dimer levels [15, 16] obtained on admission predicted mortality. Accurate body weights to determine morbid obesity were often not recorded in the hospital records of our patients, making this analysis problematic. The non-effect of D-dimer may be explained by the small size of our sample and especially since D-dimer tests were ordered by admitting physicians upon admission in only a minority of our study patients.

In other observational studies, and in contrast to our study, male sex and the presence of certain co-morbid factors, namely coronary artery disease, heart failure, cardiac arrhythmia, COPD, and current smoking status, morbid obesity, and a history of cancer predicted mortality [12, 13, 15]. Other than for atrial fibrillation, we were not able to replicate these findings, possibly because of our small sample size.

Of note we were not able to demonstrate that race or ethnicity were a significant predictors of in-hospital mortality. This is somewhat surprising as the data from the Department of Health in our state in general and Hartford County in particular, indicate that Blacks have consistently had higher mortality than Caucasians when expressed as a rate per 100,000 individuals, and Latinos fall somewhere in-between. Part of this inconsistency may be explained that the average age Caucasians in our study was about 10 years higher than that of Blacks, and age is an important predictor of bad outcome with COVID-19. Additionally, our one marker for lower SES, Medicaid or no insurance, did not predict mortality. Thus it appears that, once patients are hospitalized race/ethnicity and SES risk factors do not appear to be important predictors of mortality outcome.

Our results show some similarity and differences to those from a recent prospective, observational study performed of hospitalized COVID-19 patients in England, Scotland and Wales which was also designed to create a pragmatic risk score for in-hospital, any-cause mortality [17]. Using a very large data-set with 30.1% mortality, investigators identified 8 variables available at initial assessment and used them to create a prognostic score: age, sex, number of co-morbidities, respiratory rate, oxygen saturation, level of consciousness, blood urea nitrogen (BUN) level, and CRP.

Of note, the variables, sex, co-morbidities (other than atrial fibrillation in the preliminary analysis), respiratory rate, level of consciousness, and BUN did not prove statistically significant in our study. Age and oxygenation status were similar in the two scoring systems, since in likelihood the highest level of supplemental oxygen saturation (chosen by clinicians in the first 24 hours of hospitalization) probably reflected oxygen saturation. While level of consciousness was not included in our analysis, the reason or reasons for non-significance of other variables are not clear, although the reduced power from our considerably smaller sample probably is an important factor. Nevertheless the ROC, representing a trade-off between sensitivity and specificity, from our 4-point scoring system (0.74) was only slightly less robust than from the 22 point scoring system (0.79). Arguably, the reduction in predictive power from using fewer independent variables might be offset by its simplicity and ease of use.

A retrospective study of 403 adult patients seen in the Emergency Department in a combined secondary/tertiary care center in the Netherlands for the first wave of the pandemic (March through May, 2020) tested 11 prediction models of 30-day mortality as the primary outcome. [18]. The investigators identified two prediction models that performed best: 1) RISE-UP (acronym for Risk Stratification in the Emergency Department in Acutely Ill Older Patients) score, which included age, heart rate, mean arterial pressure, respiratory rate, oxygen saturation, Glasgow Coma Scale (GCS), BUN, bilirubin, albumin, and lactate dehydrogenase; and 2) 4-C (Coronavirus Clinical Characterization Consortium) score, which had been tested previously in the United Kingdom [19], and included age, sex, co-morbidity, RR, GCS, $O_2$ saturation, BUN, and CRP. With an AUC of 0.83 and 0.84, respectively, both performed better than ours, but were obviously more complicated in that they required entering more variables.

Comparing our results with those from other studies of in-hospital mortality is problematic for several reasons, including potential selection biases among the studies, expected regional differences in patient demographics and treatment approaches, and changes over time in therapeutic modalities for this disease.

Our study has several limitations. First is the relatively small number of hospitalized individuals studied in one medical center over the two months coinciding with peak incidence, hospitalizations and mortality in our geographical area. This obviously limits the generalizability of our results. An analysis including a larger sample (or population) of hospitalized patients performed over other health care systems—especially now that it appears mortality from this disease is changing–would be necessary to confirm these results. Since demographics of hospitalized COVID patients and in-hospital treatments are changing over time—and may be considerably different than in April and May, 2020, the time period over which we selected our study patients–the utility of our predictive models and should be tested in more contemporary patients hospitalized with COVID-19.

Another limitation of our study is the fact that some laboratory tests that proved to be predictive of mortality risk, such as CRP and troponin, were not uniformly ordered by clinicians, thereby lowering the power of our logistic regression analyses and potentially introducing a selection bias.

Finally, and similar to the study performed in the United Kingdom described above, the analysis was only of in-hospital mortality, and does not pick up mortality in patients

discharged to home, extended care facilities or hospice units. Additionally, a system that would predict in-hospital resources, not just mortality, would prove useful to hospital staff.

In summary, our results suggest that two prognostic models for hospitalized COVID-19 patients predict in-hospital mortality. Model B, consisting of three dichotomous variables present in the first 24 hours of hospitalization (age > 65 years, supplemental oxygen requirement ≥ 4 L/min, and CRP levels > 10 mg/L) may be the better of the two models tested, yielding a moderate AUC and a more robust separation of mortality between the highest and lowest scores. While the multi-component scoring models described earlier out-performed ours with respect to AUC, the simplicity of our model may prove attractive for busy emergency department staff. Our study should be viewed as presenting preliminary data which would be needed to be followed by external validation from analysis of a larger dataset across different institutions and over a longer time range, not only to attempt to replicate the findings of the predictive models, but also to determine if the scoring results in meaningful clinical consequences.

## Author Contributions

**Conceptualization:** Rahul Mutneja, Richard Zu Wallack.

**Data curation:** Nasheena Jiwa, Lucie Henry, Garrett Fiscus.

**Formal analysis:** Richard Zu Wallack.

**Methodology:** Rahul Mutneja.

**Supervision:** Rahul Mutneja, Richard Zu Wallack.

**Writing – original draft:** Nasheena Jiwa, Richard Zu Wallack.

**Writing – review & editing:** Nasheena Jiwa, Rahul Mutneja, Richard Zu Wallack.

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
