## [Decision Letter · Decision Letter 0]

5 May 2021

PONE-D-21-06850

Development of a Brief Scoring System to Predict Any-Cause Mortality in Patients Hospitalized with COVID-19 Infection

PLOS ONE

Dear Dr. Jiwa,

Thank you for submitting your manuscript to PLOS ONE. After careful consideration, we feel that it has merit but does not fully meet PLOS ONE’s publication criteria as it currently stands. Therefore, we invite you to submit a revised version of the manuscript that addresses the points raised during the review process.

We look forward to receiving your revised manuscript.

Kind regards,

Aleksandar R. Zivkovic

Academic Editor

PLOS ONE

Journal Requirements:

2) Thank you for including your ethics statement:   "Institutional Board approval was obtained prior to study initiation."

3) PLOS requires an ORCID iD for the corresponding author in Editorial Manager on papers submitted after December 6th, 2016. Please ensure that you have an ORCID iD and that it is validated in Editorial Manager. To do this, go to ‘Update my Information’ (in the upper left-hand corner of the main menu), and click on the Fetch/Validate link next to the ORCID field. This will take you to the ORCID site and allow you to create a new iD or authenticate a pre-existing iD in Editorial Manager. Please see the following video for instructions on linking an ORCID iD to your Editorial Manager account: https://www.youtube.com/watch?v=_xcclfuvtxQ

4)  Thank you for stating the following financial disclosure:

 [NO].

5) Thank you for stating the following in your Competing Interests section: 

[NO].

6) Please ensure that you refer to Figure 1 in your text as, if accepted, production will need this reference to link the reader to the figure.

Reviewers' comments:

Reviewer #1: Review “Development of a brief scoring system to predict any-cause mortality in patients hospitalized with COVID-19 infection”

First of all, I would like to thank you for the opportunity to read and review your manuscript. I have recently also been involved in research into prediction of short-term mortality in patients with COVID-19. I find the subject of your manuscript very interesting, but there are some mayor aspects that still deserve attention. Please address my comments and suggestions I have listed below.

General remarks

• As for the methodological aspects of the manuscript, I have some mayor concerns regarding the patient selection (see below).

• Abbreviations are not always sufficiently explained throughout the manuscript. For example, in the abstract, the abbreviation AUC is not spelled out. In a general sense, any abbreviation must be written out the first time it is used.

Abstract

I believe it would be better to present the abstract in the fixed structure of Introduction (or background), Methods, Results and Conclusion).

Introduction

• In line 3, the word “striking” is used twice in once sentence. I would suggest that it is removed the first time.

• The recent systematic review by Wynants et al (Reference 7) describes a relatively large number of (diagnostic) and prognostic prediction models for patients with COVID-19. In a recent publication (Performance of prediction models for short-term outcome in COVID-19 patients in the emergency department: a retrospective study. Annals of Medicine 2021;53(1):402-209), we analyzed and externally validated 11 prediction models. In the introduction of your manuscript it should be made clear why there is a need for another prediction model (i.e. what does this model add?).

Methods

• I have some mayor concerns about the selection of patients. Out of approximately 1600 patients, you selected 100 patients for the derivation cohort and 148 patients for the validation cohort (i.e. 1450 patients were not analyzed). In what way were the patients selected? How can you rule out selection bias? I think this is a very important aspect of your manuscript. Why only develop the scoring systems in 100 patients (and not more), and why validate the scores in only 148 patients (and not more)?

• You performed a retrospective study with two components. For the first component, a sample of 100 patients (out of approximately 1600 patients) was used to develop two scoring systems. In de the second component, those two scoring systems were tested. Instead of naming these samples “sample 1” and “sample 2”, I believe it would be better to refer to these samples as derivation cohort and validation cohort.

• In line 5, you state that the scoring systems were tested prospectively. I understand the scoring systems are tested (validated) on retrospective data. The word “prospectively” is confusing and should be avoided here.

• In what setting were the patients included? Emergency department or other?

• Page 4: for readers not living in the USA (like myself), the surrogate marker for low SES should be explained in more detail (Medicaid, dual Medicare-Medicaid).

• Page 6: the abbreviation ABR should be changed to ARB (Angiotensin receptor blocker).

• The section on statistical analysis should have its own subheading.

Results

• As mentioned above, I would suggest the two study samples are referred to as derivation cohort and validation cohort.

• The text belonging to subheadings “Sample 1” and “Sample 2” are actually a summary of the data from Table 1. Consider merging the two subheading into one (e.g. “Study samples”) and shortening the text to just the most important data.

• Page 6 (Subheading “Sample 2”): in this section you described why you ended up with 148 patients in the validation cohort. I believe this explanation should be moved to the Methods section. In my opinion, you should also clarify whether different patients were analyzed in both cohorts, or whether the two cohort show an overlap. The reason for inclusion or exclusion of patients in the cohorts could also be explained in further detail.

• It would greatly improve your manuscript if you could show the patient characteristics of the total population of 1600 patients. This allows the reader to estimate any selection bias. I would suggest adding an extra Table with this information.

• Page 6: The subheading “Creating in-hospital mortality prognostic scoring models from Sample 1 cohort” is too long. It could be changed to “Selection of predictors and univariate and multivariate logistic regression analysis”.

• Page 7: In line 4, “Hispanic mortality” is misspelled as “Hispanic morality”. The point after 40% should be removed.

• Page 8: The subheading “Testing the models using Sample 2 data” could be changed to “Validation of the scoring systems”.

• Page 8: How are the two ROC curves compared? This statistical analysis should be explained in more details in the statistical analysis section.

• Page 8: Model A could be used in only 107 of 148 patients, because of missing data regarding troponin (27%). Since troponin is not routinely measured, is model A feasible in clinical practice?

Discussion

• Page 9: You recommend the use of Model B in patients with COVID-19 for predicting in-hospital mortality. However, your analysis shows no statistically significant difference between model A and B (again: the method of comparing the two models is unclear). So, why not use model A?

• Page 9: What are the clinical consequences of the use of your scoring systems? In other words, should a low or a high score guide clinical decision-making?

• Page 9: The sentence “For reference, an AUC between … considered excellent” belongs to the Methods section and should not be part of the Discussion section.

• Page 10: Before being able to compare your results with those from other studies, it is important to estimate the degree of selection bias. See previous comments.

• Page 10-11: In addition to the British 4C mortality score (Knight and colleagues), the RISE UP score is also very useful for predicting short term adverse outcome in ED patients with COVID-19 (Annals of Medicine 2021;53(1):402-209 and BMJ Open 2021;11:e045141).

---

## [Author Response · Author response to Decision Letter 0]

27 Jun 2021

Dear PLOS ONE Editorial Staff,

We want to thank you and your reviewers for your thorough review. The text below outlines the reviewers’ comments/questions and our responses.

We wish to re-submit the revised manuscript for consideration for publication.

Sincerely,

Nasheena Jiwa, MD

Journal Requirements:

The Trinity Health of New England Institutional Review Board granted approval prior to study initiation

Corresponding Author’s ORCID iD: https://orcid.org/0000-0003-0261-2425

The authors received no specific funding for this work.

The authors have declared that no competing interests exist.

Please ensure that you refer to Figure 1 in your text as, if accepted, production will need this reference to link the reader to the figure.

Figure 1 is now referenced in the text in the last paragraph of the Discussion.

Reviewer Comments:

General Remarks

- First occurrence of area under the curve (AUC) is now spelled out in the Methods section

Abstract: This was revised to reflect the comments by the reviewer:

Abstract 

Patients hospitalized with COVID-19 infection are at a high general risk for in-hospital mortality. A simple and easy-to-use model for predicting mortality based on data readily available to clinicians in the first 24 hours of hospital admission might be useful in directing scarce medical and personnel resources toward those patients at greater risk of dying. With this goal in mind, we evaluated factors predictive of in-hospital mortality in a random sample of 100 patients (derivation cohort) hospitalized for COVID-19 at our institution in April and May, 2020 and created potential models to test in a second random sample of 148 patients (validation cohort) hospitalized for the same disease over the same time period in the same institution. Two models (Model A: two variables, presence of pneumonia and ischemia); (Model B: three variables, age > 65 years, supplemental oxygen ≥ 4 L/min, and C-reactive protein (CRP) > 10 mg/L) were selected and tested in the validation cohort. Model B appeared the better of the two, with an AUC in receiver operating characteristic curve analysis of 0.74 versus 0.65 in Model A, but the AUC differences were not significant (p = 0.24. Model B also appeared to have a more robust separation of mortality between the lowest (none of the three variables present) and highest (all three variables present) scores at 0% and 71%, respectively. These brief scoring systems may prove to be useful to clinicians in assigning mortality risk in hospitalized patients.

Introduction:

- Deleted “striking” 

- Our prediction model is intended to be a used as a tool for quick interpretation of patient data within the first 24 hours, to predict mortality outcomes

Methods:

In what way were the patients selected? How can you rule out selection bias? I think this is a very important aspect of your manuscript. Why only develop the scoring systems in 100 patients (and not more), and why validate the scores in only 148 patients (and not more)?

Patients in both samples were randomly selected from the pool of patients hospitalized with COVID-19. 

This is stated in Methods as follows: 

“For the first component of the study, we reviewed in-hospital records from a randomized sample of 100 COVID-19 patients (out of approximately 1600 records), who had been admitted to our tertiary-care center in Hartford, CT, during the months of April and May, 2020. These patients are known as the derivation cohort. All patient’s had been admitted and hospitalized with a clinical diagnosis and serological confirmation of COVID-19 infection. 

“A second sample of different patients called the validation cohort of patients was then reviewed. These records were randomly selected from the same population of hospitalized patients in April and May, 2020 (excluding those in the original sample), until an arbitrary number of at least 40 in-hospital deaths from COVID-19 were reviewed.” 

Because of limitations in research staff (this study was not externally funded) we could not study a larger sample. However, it should be noted that we did achieve statistical significance in several predictor variables that also appear mechanistically sound. 

Corrected sample 1 = derivation cohort, and sample 2 = validation cohort

Removed the word “prospectively”

Patient’s included were admitted to the hospital 

Explained in detail: low SES (Medicaid, dual Medicare-Medicaid).

abbreviation ABR was changed to ARB (Angiotensin receptor blocker).

Added a statistical analysis subheading

Results

Corrected sample 1 = derivation cohort, and sample 2 = validation cohort

Description of sample 2/validation cohort of patients was moved to methods section

Clarified that both cohorts contained different patients 

It would greatly improve your manuscript if you could show the patient characteristics of the total population of 1600 patients. This allows the reader to estimate any selection bias. I would suggest adding an extra Table with this information.

As we stated earlier, we did not have the research personnel to do a chart review on all 1600 patients. However, we could note that the two random samples were similar with respect to patient characteristics, as given in Table 1. 

The subheading “Creating in-hospital mortality prognostic scoring models from Sample 1 cohort” was changed to “Selection of predictors and univariate and multivariate logistic regression analysis”

Corrected: “Hispanic mortality” is misspelled as “Hispanic morality”. 

“Testing the models using Sample 2 data” was changed to “Validation of the scoring systems”.

How are the two ROC curves compared? This statistical analysis should be explained in more details in the statistical analysis section.

We clarified this with the following added text: “ROC curves were compared using the SAS logistic procedure and the ROCCONTRAST statement.”

- 

Model A could be used in only 107 of 148 patients, because of missing data regarding troponin (27%). Since troponin is not routinely measured, is model A feasible in clinical practice?

We changed the text in the Discussion to reflect this appropriate comment as follows: “Based on performance in the validation cohort, Model B had a slightly higher AUC (0.74 vs. 0.65), although the difference between the two models was not statistically significant. Model B also tended to perform better with respect to mortality separation in logistic regression. Given the small numbers of subjects in our study, the lack of a statistically significant difference in AUC, and the fact that CRP (a component of model B) was not obtained in all patients upon admission, a strong inference on the comparative performance of the two scoring systems would be problematic.”

Discussion:

You recommend the use of Model B in patients with COVID-19 for predicting in-hospital mortality. However, your analysis shows no statistically significant difference between model A and B (again: the method of comparing the two models is unclear). So, why not use model A? 

This was addressed with the changes in text above in the Discussion:

What are the clinical consequences of the use of your scoring systems? In other words, should a low or a high score guide clinical decision-making

The text in the Discussion was changed to reflect the uncertainty of the clinical consequences: 

“Our study should be viewed as presenting preliminary data which would be needed to be followed by external validation from analysis of a larger dataset across different institutions and over a longer time range, not only to attempt to replicate the findings of the predictive models, but also to determine if the scoring results in meaningful clinical consequences.”

Removed “For reference, an AUC between … considered excellent” from the discussion section and moved it to the Methods section 

Page 10: Before being able to compare your results with those from other studies, it is important to estimate the degree of selection bias. See previous comments.

We added the following disclaimer in the Discussion: 

“Comparing our results with those from other studies of in-hospital mortality is problematic for several reasons, including potential selection biases among the studies, expected regional differences in patient demographics and treatment approaches, and changes over time in therapeutic modalities for this disease.” 

Page 10-11: In addition to the British 4C mortality score (Knight and colleagues), the RISE UP score is also very useful for predicting short term adverse outcome in ED patients with COVID-19 (Annals of Medicine 2021;53(1):402-209 and BMJ Open 2021;11:e045141).

These studies were added to the Discussion and referenced. Thank you.

“A retrospective study of 403 adult patients seen in the Emergency Department in a combined secondary/tertiary care center in the Netherlands for the first wave of the pandemic (March through May, 2020) tested 11 prediction models of 30-day mortality as the primary outcome. (18) The investigators identified two prediction models that performed best: 1) RISE-UP (acronym for Risk Stratification in the Emergency Department in Acutely Ill Older Patients) score, which included age, heart rate, mean arterial pressure, respiratory rate, oxygen saturation, Glasgow Coma Scale (GCS), BUN, bilirubin, albumin, and lactate dehydrogenase; and 2) 4-C (Coronavirus Clinical Characterisation Consortium) score, which had been tested previously in the United Kingdom, (19) and included age, sex, co-morbidity, RR, GCS, O2 saturation, BUN, and CRP. With an AUC of 0.83 and 0.84, respectively, both preformed better than ours, but were obviously more complicated in that they required entering more variables.”

---

## [Editor Report · Decision Letter 1]

30 Jun 2021

Development of a Brief Scoring System to Predict Any-Cause Mortality in Patients Hospitalized with COVID-19 Infection

PONE-D-21-06850R1

Dear Dr. Jiwa,

We’re pleased to inform you that your manuscript has been judged scientifically suitable for publication and will be formally accepted for publication once it meets all outstanding technical requirements.

Kind regards,

Aleksandar R. Zivkovic

Academic Editor

PLOS ONE

---

## [Editor Report · Acceptance letter]

8 Jul 2021

PONE-D-21-06850R1 

Development of a Brief Scoring System to Predict Any-Cause Mortality in Patients Hospitalized with COVID-19 Infection 

Dear Dr. Jiwa:

I'm pleased to inform you that your manuscript has been deemed suitable for publication in PLOS ONE. Congratulations! Your manuscript is now with our production department. 

Kind regards, 

on behalf of

Dr. Aleksandar R. Zivkovic 

Academic Editor

PLOS ONE